# Boosted Trees Algorithm as Reliable Spectrum Sensing Scheme in the Presence of Malicious Users

**Noor Gul [1,2], Muhammad Sajjad Khan [1,3] 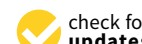, Su Min Kim [3], Junsu Kim [3,*], Atif Elahi [1] and Zafar Khalil [4]**

[1] Department of Electrical Engineering, Faculty of Engineering and Technology, International Islamic University, Islamabad 44000, Pakistan; noor.phdee51@iiu.edu.pk (N.G.); sajjad.khan@iiu.edu.pk (M.S.K.); atif.phdee40@iiu.edu.pk (A.E.)

[2] Department of Electronics, University of Peshawar, Peshawar 25120, Pakistan

[3] Department of Electronics Engineering, Korea Polytechnic University, Gyeonggi-do 15073, Korea; suminkim@kpu.ac.kr

[4] Department of Computer Science, Northern University, Nowshera 24110, Pakistan; zafarkhan@northern.edu.pk

\* Correspondence: junsukim@kpu.ac.kr; Tel.: +82-31-8041-0497

**Abstract:** Cooperative spectrum sensing (CSS) has the ability to accurately identify the activities of the primary users (PUs). As the secondary users' (SUs) sensing performance is disturbed in the fading and shadowing environment, therefore the CSS is a suitable choice to achieve better sensing results compared to individual sensing. One of the problems in the CSS occurs due to the participation of malicious users (MUs) that report false sensing data to the fusion center (FC) to misguide the FC's decision about the PUs' activity. Out of the different categories of MUs, Always Yes (AY), Always No (AN), Always Opposite (AO) and Random Opposite (RO) are of high interest these days in the literature. Recently, high sensing performance for the CSS can be achieved using machine learning techniques. In this paper, boosted trees algorithm (BTA) has been proposed for obtaining reliable identification of the PU channel, where the SUs can access the PU channel opportunistically with minimum disturbances to the licensee. The proposed BTA mitigates the spectrum sensing data falsification (SSDF) effects of the AY, AN, AO and RO categories of the MUs. BTA is an ensemble method for solving spectrum sensing problems using different classifiers. It boosts the performance of some weak classifiers in the combination by giving higher weights to the weak classifiers' sensing decisions. Simulation results verify the performance improvement by the proposed algorithm compared to the existing techniques such as genetic algorithm soft decision fusion (GASDF), particle swarm optimization soft decision fusion (PSOSDF), maximum gain combination soft decision fusion (MGCSDF) and count hard decision fusion (CHDF). The experimental setup is conducted at different levels of the signal-to-noise ratios (SNRs), total number of cooperative users and sensing samples that show minimum error probability results for the proposed scheme.

**Keywords:** cognitive radio; machine learning; genetic algorithm; cooperative communication; particle swarm optimization; boosted trees algorithm; multipath channels

## 1. Introduction

The tremendous growth in wireless communication technology has resulted in a shortage of the radio spectrum in the last few decades. Therefore, it was suggested by the Federal Communication Commission (FCC) to follow more flexible and comprehensive use of the available spectrum resources using cognitive radio (CR) technology to overcome this issue [1,2]. The spectrum measurement study

made in different parts of the world shows spectrum access is a more serious problem than physical inadequacy [2] as spectrum resources most of the time are either partially in use of the licensed users or fully unoccupied. Therefore, dynamic spectrum access (DSA) is needed that enables flexible management policy of the spectrum resources. DSA allows unrestricted and dynamic access to different spectral resources from the users operating in a similar frequency range. Therefore, the bodies are heading towards DSA and rejecting the static access methods to resolve the spectrum scarceness.

The CR idea was presented for the first time by Mitola with derivation from the Latin word "cognoscere" meaning to come to know or to become aware of something [3]. This provides a new method for wireless communication design with the objective to provide wireless communication services using the DSA method. CR has an ability to improve the performance of wireless communication by overcoming frequency spectrum consumption that leads to reduced spectrum underutilization problems. The duty cycle of the CR consists of sensing, analysis reasoning and adaptation. Sensing refers to the detection of the primary user (PU) spectral holes at the secondary users (SUs) and considered to be one of the main functional components of the CR network [4,5]. In the analysis, CR has to select the best available frequency band and coordinates the same with other CR neighbors. At last, spectrum resources are vacated whenever the PU reappears and starts transmission in the given spectrum band. The spectrum sensing job is more critical out of them in order to avoid disturbances created by the SUs for the licensed PUs. As the multipath fading and shadowing effects drastically reduce the performance of the individual sensing, and therefore, cooperative spectrum sensing (CSS) helps in reducing these sensing difficulties [6]. The two broad categories of CSS are centralized and distributed sensing mechanisms. Centralized CSS enables cooperative users to report their individual sensing observations to the fusion center (FC) for making the final global decision. In the distributed CSS, individual users share sensing information with each other without reporting the same to any central control station [7,8]. Distributed fusion algorithms in [9] affect collaborative fusion at low communication and computation loads scalable to the network size. The focus in [10] is on the local sensing using OFDM-based PU transmission with CSS using sequential and censoring schemes.

The security issues in the CSS cannot be ignored despite its considerable advantages. Investigation of malicious users (MUs) to secure the CSS against the MUs' attack is a challenging area that has received researchers' attention. The attacker's job is to confuse FC about the actual usage of the PU spectrum resources to get the benefit of the spectrum for its own usage. It is, therefore, necessary to provide protection to the CSS against these categories of spectrum falsification attackers [11–16].

The most commonly used sensing detectors for CSS in most of the literature are the matched filter detector (MFD), energy detector and feature detectors. The prior knowledge requirements of the PU channel make MFD difficult to implement. Energy detectors are easy to implement in the CSS due to simple hardware requirements at the sensing user's end.

The CSS is vulnerable to security threats from malicious nodes that alter PU channel sensing information for personal benefit. The work in [17] quantified the number of affected nodes due to the malicious sensing reports in simulation scenarios of the Poisson point process. Out of the different categories of attacker Byzantine users, jammers and primary user emulation attackers (PUEA) severely affect the CSS performance. In Byzantine attacks, the MUs report false sensing data to the FC [18]. Similarly, in the jamming attack, the received signal at the receiver is interfered with by the MUs. PUEA attackers try to masquerade as a legitimate PU by preventing other users' access to the spectrum [19].

In [20], security against Byzantine users is suggested using a novel corruption implementation strategy to help FC. This scheme allows the FC to distinguish attackers and normally cooperative users using the message passing algorithm [20]. As the attacking MUs are not willing to share honest sensing information with the FC, a contract theory approach is proposed in [21] as an incentive design mechanism. This scheme rewards honest cooperative users by strengthening their future cooperation. In [22], a novel algorithm is proposed to deal with an arbitrary number of Byzantine attackers. A clean data set is selected in this work by the tuning parameter server that computes noise gradient to filter compromised participants' information. The abnormal power emission due to faulty devices, selfish

motivation and malicious reports are formulated as a composite binary hypothesis test in [23]. Location reliability and malicious intention (LRMI) parameters in [24] improve the detection of both malicious and mobile CRNs. A recursive updating algorithm and order statistics proposed in [25] helps the SUs selection with a higher sensing reputation to reduce MUs impact on CSS. The honest SUs in [26] follow the FC decision as a final recommendation of the PU channel and uses its local sensing reports to guarantee the CSS reliability. A privacy-preserving protocol in [27] uses cryptographic schemes to preserve SU location privacy with maintaining the reliability of the sensing. A privacy-preserving scheme using an additional architectural entity along with cryptographic techniques maintains user location privacy, while performing spectrum sensing [28]. Similarly, cooperative users' data falsification effects are reduced by measuring and updating the sensing users' credit in [29]. An efficient data fusion scheme is investigated in [30] that counters the effects of the SSDF attackers in CSS. This leads to minimum security issues in the cognitive radio wireless sensor network at the participation of PUEA and SSDF attackers. A novel attack proof method for CSS including MU identification with adaptive linear combination technique is discussed in [31].

Individual attackers sometimes collude to create a stronger attacking strategy at the FC. As the collusion attackers have the capability to enhance the individual attackers' strength, therefore, they are able to easily avoid defense mechanisms employed at the FC. A secure defense scheme from the XOR distance measurement perspective is proposed in [32] that suppresses collusion SSDF attackers in CSS. A problem with the trust-based CSS feedback scheme is the data initiator SUs are unchecked at the FC. The spectrum information of the PU status feedback by FC is often exploited by attackers to disturb the trust mechanism in a collusive manner [33]. A Bayesian inference scheme following the sliding window trust model identifies and reduces the effects of the individual and probabilistic collaborative SSDF attackers in [34]. The system security is enhanced in the penalty-based collision prevention scheme in [35] at the contribution of different types of individual and collaborative attackers. A dynamic collusive SSDF attacker in [36] maintains high-level trust at the FC with reporting accurate and fake sensing data dynamically to strengthen attackers' behavior. Similarly, the performance of the randomly selected false category of attack is discussed under the individual and collaborative attackers at the FC in [37]. As most of the security schemes assume attacker patterns at the FC, therefore the work in [38] adopted an arbitrary behavior that restricts assumed patterns and assumptions of the attacker using the learn–evaluate–beat (LEB) method.

A deep cooperative sensing (DCS)-based spectrum sensing solution for CRNs is investigated in [39]. The use of the K-nearest neighbor (KNN) is proposed in [40–42] to solve spectrum sensing problems. The scheme proposed in [43] is following extreme machine learning to get reliable channel information for the SUs, in order to temporarily share a channel with the PUs. A solution of the machine-learning-based trust (MLBT) is proposed in [44] to detect malicious activities in the vehicular-based machine-to-machine communication (VBM2M-C). Machine-learning- and statistical-analysis-based approach in [45] detect malicious software in both mobile devices and caters. CSS performance based on machine learning is recently considered as a frontier in CRNs. The work in [46] pointed out that CSS in combination with machine learning boosts decision capability by learning from past experience. Therefore, it allows us to make use of the unstable opportunistic spectrum resources more effectively than traditional ways of using the spectrum. A reinforcement learning scheme is proposed in [47] to improve the sensing performance of the individual sensing node. The use of KNN and supervised machine learning (SVM) for CSS is suggested in [48]. A supervised and unsupervised classification technique that classifies the channel into available and unavailable classes is discussed in [49]. The use of an artificial neural network (ANN) in combination with a cyclostationary-based sensing scheme is proposed in [50]. Similarly, a fusion scheme based on pattern recognition for spectrum sensing is proposed in [51]. The use of the sample covariance matrix in machine-learning-based spectrum sensing is proposed in [52]. The balance in complexity and performance is achieved using novel CSS at the FC by following ANN in [50]. Similarly, an unsupervised machine learning technique with better sensing performance is investigated in [53].

In our previous work in [54] and [55], statistical Kullback–Leibler (KL) divergence schemes are employed. Similarly to [56,57], we employed a genetic algorithm (GA), to govern suitable PU-sensing information for a global decision based on the reported hard binary data of the users. In our outlier detection method in [58], we used one-to-many sensing distance and Z-score statistical results of the user's hard binary decisions.

In this paper, we proposed a novel CSS based on machine learning techniques using a boosted tree algorithm (BTA) with AdaBoost as an ensemble method. Ensemble learning is a useful method to achieve more detection accuracy compared with the detection performance of the individual classifiers. Depression detection and intrusion detection schemes are analyzed in abnormal echo propagation for weather radar, which are investigated in [59,60]. A smart grid environment using AdaBoost for islanding detection is proposed in [61]. This paper is the first paper applying BTA to the CSS in CRN, when MUs report false sensing data to the FC. The BTA scheme proposed in this paper for CSS leads to better sensing results at the FC with less impact and disturbance from the different categories of MUs. In our proposed work, a feature vector is extracted from patterns and fed into the classifier to categorize the given pattern into a class with the most relevant suitability. In the CSS environment, our feature vector consists of sensing energies that show statistics of the individual cooperative users while observing the PU channel. Then, the boosted trees classifier tries to categorize the input energy vector into one of the two classes such as the PU channel available class (no PU activity in the given spectrum) and the PU channel busy classes (PU activity in the given spectrum). Before actual classification of the real-time sensed energy vector, the proposed boosted tree algorithm (BTA) passes through the training phase, where it learns from the energy feature vector input into the system.

This scheme has an ability to improve CSS performance in the presence of Always Yes (AY), Always No (AN), Always Opposite (AO) and Random Opposite (RO) categories of MUs along with their collusion attacks. The AY user reports their malicious high energy-sensing observations to the central FC and reports the same to their Always Yes Collusion (AYC) center. AN users inform the main FC and Always No Collusion (ANC) with information that always negates the availability of the PU channel. Similarly, the RO user always informs the main FC with sensing information that is in negation with the actual status of the PU channel and reports this information to the Random Opposite Collusion (ROC) center. The system is trained using the BTA that uses AdaBoost as an ensemble method. The training data set is formed based on sensing observations of the normally behaving users, AY, AN, AO, AYC, ANC and Always Opposite Collusion (AOC) categories of malicious attacks. The performance of the proposed boosted tree scheme is further examined under the testing sensing observations reported by all cooperative users in CSS.

The results demonstrate reliable and accurate detection performance of the PU channel by the proposed BTA-based CSS as compared to the particle swarm optimization soft decision fusion (PSOSDF), genetic algorithm soft decision fusion (GASDF), count hard decision fusion (CHDF) and maximum gain combination soft decision fusion (MGCSDF) schemes in [14,54,62]. The simulation environment is categorized into four different cases including normal users in cooperation, AY/AYC users, AN/ANC users and AO/AOC users' contributions in the cooperative environment. Error probability results are further collected against the varying signal-to-noise ratios (SNRs), cooperative users and sensing samples that confirmed minimum error probability, high detection probability and low false alarm results for the proposed BTA-based scheme.

The paper is further categorized into the following sections. The system model is discussed in Section 2. The proposed model for determining optimal weighted results using GA is in Section 3. Section 4 illustrates the simulation outcomes. Finally, concluded remarks and further research directions are included in Section 5.

## 2. System Model

In the conventional CSS model in Figure 1, users sense the PU channel and report their statistics to the FC. The normal users in this scheme report the actually measured information of the channel to the FC, while the malicious users report false sensing information to mislead the FC. Figure 2 shows the energy distributions of the normal and different types of malicious users. It senses the PU channel and reports low energy statistics if the channel is busy and likewise forwards high energy statistics in the absence of PU. The presence of AO/AOC users in the CSS leads to a reduced data rate and increased interference with the PU. The MUs in the AY/AYC category always report high energy statistics, which lead to a reduced data rate to the SUs by creating false alarms. MUs of AN/ANC category sense the PU activity and report the FC with low energy statistics to present always idle conditions of the channel that results in interference for the PU. Similarly, the RO/ROC users probabilistically operate equally as AO with probability (P) and operate normally with probability (1-P). The involvement of these MUs in the CSS leads the FC to decide inaccurately about the PU activity, therefore they can reduce the data rate and create unacceptable interference for the legitimate users.

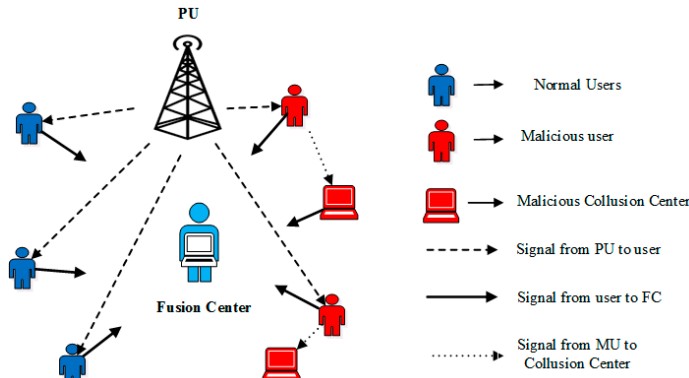

**Figure 1.** Conventional centralized cooperative spectrum sensing (CSS).

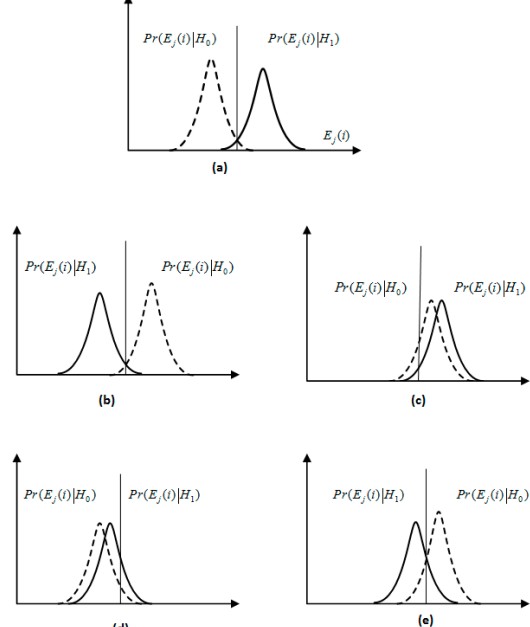

**Figure 2.** The PDF of the normal and malicious sensing reports: (**a**) Normal, (**b**) Always Opposite (AO)/Always Opposite Collusion (AOC), (**c**) Always Yes (AY)/Always Yes Collusion (AYC), (**d**) Always No (AN)/Always No Collusion (ANC) and (**e**) Random Opposite (RO)/Random Opposite Collusion (ROC).

In centralized CSS networks, cognitive users report their soft energy observations of the channel to the FC for making a more suitable global decision using SDF in the fading and shadowing environment. CSS is investigated in this work under the contribution of normal users along with AY, AN, AO, AYC, ANC and AOC categories of MUs.

The $H_0$ and $H_1$ hypothesis for the $j^{th}$ user at time slot $l$ of the PU channel are as [8]

$$\begin{cases} H_0 : x_j(l) = v_j(l) \\ H_1 : x_j(l) = g_j c(l) + v_j(l) \end{cases}, \ j \in \{1, 2, \ldots, M\}, l \in \{1, 2, \ldots, K\}, \tag{1}$$

where $x_j(l)$ is the PU received signal at the $j^{th}$ user out of $M$ total users in the $l^{th}$ sensing time slot. $H_0$ shows no activity of the PU in the allocated spectrum, while $H_1$ hypothesis is to represent the busy status of the channel that is not for the use of opportunistic SUs. The sensing energy is Gaussian under a sufficient number of sensing samples $K = 2BT_s$ under both $H_0$ and $H_1$ hypothesis, where B is the bandwidth and $T_s$ is the sensing period. Similarly, the gain of the channel between PU and $j^{th}$ sensing user is $g_j.c(l)$ represent PU channel sensing samples in the $l^{th}$ sensing slot with variance $\sigma_s^2$ and mean zero. $v_j(l)$ is additive white gaussian noise (AWGN) at the channel between the $j^{th}$ user and PU with mean zero and variance $\sigma_{v_j}^2$.

The observed energy based on the $H_1$ and $H_0$ in (1) is represented as [11]

$$s_j(i) = \begin{cases} \sum_{l=l_i}^{l_i+K-1} |v_j(l)|^2, & H_0 \\ \sum_{l=l_i}^{l_i+K-1} |g_j c(l) + v_j(l)|^2, & H_1 \end{cases}, \tag{2}$$

where each sensing interval is divided into $K$ total samples. In the case of a sufficient number of sensing samples, the soft energy reports of these SUs converges to a Gaussian random variable in both $H_0$ and $H_1$ hypothesis as [11]

$$s_j \sim \begin{cases} N\left(\mu_0 = K, \sigma_0^2 = 2K\right), & H_0 \\ N\left(\mu_1 = K(\eta_j + 1), \sigma_1^2 = 2K(\eta_j + 1)\right), & H_1 \end{cases}, \tag{3}$$

Here, $\eta_j$ is the SNR between the $j^{th}$ user and the PU, while $(\mu_0, \sigma_0^2)$ and $(\mu_1, \sigma_1^2)$ are energy distribution means and variances when $H_0$ and $H_1$ hypothesis is true.

## 3. Proposed Boosted Trees Algorithm

The proposed algorithm's working principle is categorized into three major steps. In Step 1, $N$ sensing observations from the normal and different categories of MUs are collected at the FC that work as a set of feature vectors for the proposed BTA scheme. Step 2 trains the BTA using the AdaBoost ensemble method. The BTA makes more precise and suitable decisions by accumulating and strengthening the weak classifiers in Step 2. Step 3 employs Step 2's results to determine the detection and false alarm probabilities based on the global decision on the PU activity.

### 3.1. Step 1

The FC receives the soft energy reports from all individual SUs to form a history reporting matrix including soft energy statistics observation of each SUs in the $i$ sensing intervals as below

$$S = \left[s_{ij}\right] = \begin{bmatrix} s_{11} & s_{12} & \cdots & s_{1M} \\ s_{21} & s_{22} & \cdots & s_{2M} \\ s_{31} & s_{32} & \cdots & s_{3M} \\ \vdots & \vdots & \ddots & \vdots \\ s_{N1} & s_{N2} & \cdots & s_{NM} \end{bmatrix}, i \in \{1, 2, \ldots, N\}, j \in \{1, 2, \ldots, M\}, \tag{4}$$

where $s_{ij}$ is the $j^{th}$ user's energy detection result as in (2) reported to the FC in the $i^{th}$ interval. Spectrum sensing data is accumulated at the FC for all $M$ SUs including normal and malicious users in the $N$ sensing intervals consists of $K$ samples.

The spectrum sensing data falsification effects of the MUs are minimized using the proposed technique in the following steps. Machine learning algorithms find natural patterns in the data that help in decision making and predictions.

*3.2. Step 2: Training Using Boosted Trees Algorithm*

The BTA is a category of ensemble classifiers that create a strong and highly accurate classifier with multiple weak classifiers that combine their outputs to improve prediction. The training dataset is used to train the set of weak classifiers to generate a combined ensemble prediction model. This allows the weak classifiers to update, hence it eliminates the need for retraining procedure. The diversity in the weak classifiers' outputs offers the advantage of achieving higher predictions, since each data type is expected to have various characteristics of the classification instance. As the individual learner/classifier tends to produce biased predictions, and therefore, ensemble strategy is a better choice to fix the biases by integrating and optimizing the poor results through certain combination schemes to obtain more accurate decisions [63].

The use of ensemble learning method is a useful way to achieve detection accuracy with more authenticity. The ensemble classifiers showed promising results in many detection problems [63–65]. Multiple learning algorithms are used by the ensemble classifiers in order to achieve higher prediction efficiency than their weak/base classifiers [66]. The ensemble classifiers scheme follows the divide and conquer strategy for solving complex problems [67]. Depression detection is investigated in speech signals using the ensemble method in [63]. A network intrusion detection scheme using an ensemble classifier is proposed in [63]. An electrocardiograph artifact is detected and analyzed using the ensemble decision trees method in [59]. In [60], abnormal propagation echo for weather radar is detected using an ensemble classifier. An ensemble classifier and AdaBoost are used for islanding detection in a smart grid environment in [61].

The BTA uses adaptive boosting (AdaBoost) as an ensemble method, where a strong classifier is constructed by assembling some weak classifiers. In this work, the training set is $T = \left\{\left(s_{ij}, y_i\right)\right\}_{i=1}^{N}$, $s_{ij} \in \mathfrak{R}^N$, $y_i \in \{-1, 1\}$, where $y_i = 1$ denotes the class label presence of the PU activity and $y_i = -1$ denotes the class label to show the absence of the PU activity. The given training set $T$ is considered as an $N \times (M+1)$ matrix with dimensional space $T \in \mathfrak{R}^{N \times (M+1)}$, which can be written as

$$T = \begin{bmatrix} \begin{array}{cccc|c} s_{11} & s_{12} & \cdots & s_{1M} & y_1 \\ s_{21} & s_{22} & \cdots & s_{2M} & y_2 \\ \vdots & \vdots & \ddots & \vdots & \vdots \\ s_{N1} & s_{N2} & \cdots & s_{NM} & y_N \end{array} \end{bmatrix},$$ (5)

where $s^1 = \begin{bmatrix} s_{11} \\ s_{12} \\ \vdots \\ s_{1M} \end{bmatrix}^T$, $s^2 = \begin{bmatrix} s_{21} \\ s_{22} \\ \vdots \\ s_{2M} \end{bmatrix}^T$, $s^3 = \begin{bmatrix} s_{31} \\ s_{32} \\ \vdots \\ s_{3M} \end{bmatrix}^T$ and $s^N = \begin{bmatrix} s_{N1} \\ s_{N2} \\ \vdots \\ s_{NM} \end{bmatrix}^T$ are the different feature vectors

that show the reported sensing data including SNRs and soft energy reports. The training set consists of two submatrices $S$ and $Y$. Here, submatrix $S \in \mathfrak{R}^{N \times M}$ is the matrix features of user sensing data and $Y \in \mathfrak{R}^{N \times 1}$ is a label matrix representing the status of the PU activity.

Here, for the case of $s^i$ as the $i^{th}$ feature vector to be classified as a decision of the presence and absence of licensee spectrum condition, total $k$ classifiers participate in the sensing decision. The different classifiers predict a decision (class) label for $s^i$ and the final output label is estimated by the linear combination of the predicted label estimation of the different classifiers. In this linear combination, each term is the product of the class label value as predicted by the classifier and the

assigned weights to the predicted values of the classifier. In AdaBoost, each $k^{th}$ classifier is assigned with decision weights by knowing the predictions of the $(k-1)$ classifiers to represent the boosted classifier as

$$e_{k-1}(s^i) = \sum_{p=1}^{k-1} \alpha_p h_p(s^i), \ p \in \{1, 2, \dots, k-1\}, \ i \in \{1, \dots, N\},$$ (6)

where $h_p(s^i)$ is the value predicted by the $p^{th}$ classifier and $\alpha_p$ is the weight assigned to the classifier predicted value.

Similarly, the $k^{th}$ classifier prediction performance is included with $h_k(s^i)$ as predicted value and $\alpha_k$ optimum weight to the classifier to form a better-boosted classifier. The measurement process for $\alpha_k$ is as follows

$$e_k(s^i) = \sum_{p=1}^{k-1} \alpha_p h_p(s^i) + \alpha_k h_k(s^i).$$ (7)

Using the value from (6), the above can be written as

$$e_k(s^i) = e_{k-1}(s^i) + \alpha_k h_k(s^i),$$ (8)

where $e_k(s^i)$ is the compound predicted value obtained by aggregating the predicted values of $k$ classifiers. We are interested in a closed-form formula for $\alpha_k$, which assigns $\alpha_k$ a value in such a way that the total error of prediction is minimized. The total error of prediction in adaptive boosting is defined as the sum of the negative natural exponential of $y^i e_k(s^i)$ for all the training examples as given below [68].

$$E = \sum_{i=1}^{n} e^{-y^i [e_{k-1}(s^i) + \alpha_k h_k(s^i)]}.$$ (9)

The total error expression now takes the form

$$E = \sum_{i=1}^{n} w_k^i e^{-y^i \alpha_k h_k(s^i)},$$ (10)

where $w_k^i = e^{-y^i e_{k-1}(s^i)}$ is the weight in the case of classifier number $k > 1$. Next, we split the error in (10) into the cases where the prediction is correct ($y^i h_k(s^i) = 1$) and where the prediction is incorrect ($y^i h_k(s^i) = -1$) as

$$\begin{aligned} E &= \sum_{y^i = h(s^i)} w_c e^{-y^i \alpha_k h_k(s^i)} + \sum_{y^i \neq h(s^i)} w_e e^{-y^i \alpha_k h_k(s^i)}, \\ &= \sum_{y^i = h(s^i)} w_c e^{-\alpha_k} + \sum_{y^i \neq h(s^i)} w_e e^{\alpha_k}, \\ &= e^{-\alpha_k} \sum_{y^i = h_k(s^i)} w_c + e^{\alpha_k} \sum_{y^i \neq h_k(s^i)} w_e, \\ &= W_c e^{-\alpha_k} + W_e e^{\alpha_k}, \end{aligned}$$ (11)

where $W_c = \sum_{y^i = h_k(s^i)} w_c$ and $W_e = \sum_{y^i \neq h_k(s^i)} w_e$. $E$ is the loss function we want to minimize for the chosen weak classifier $h_k$ selected previously. Therefore, the total error $E$ is differentiated with classifier weight $\alpha_m$ and the minimization condition is set to zero such as

$$\frac{\partial E(\alpha_k)}{\partial \alpha_k} = 0.$$ (12)

From (11), minimization of $E$ reference to $\alpha_k$ is similar to the minimization of $(W_c e^{-\alpha_k} + W_e e^{\alpha_k})$

$$\frac{\partial E(\alpha_k)}{\partial \alpha_k} = -W_c e^{-\alpha_k} + W_e e^{\alpha_k}. \tag{13}$$

The expression in (13) is solved for $\alpha_k$ as

$$\alpha_k = \frac{1}{2} \ln\left(\frac{W_c}{W_e}\right). \tag{14}$$

Because $W_c = W - W_e$, where $W$ is the total sum of the weights, therefore

$$\alpha_k = \frac{1}{2} \ln\left(\frac{W - W_e}{W_e}\right). \tag{15}$$

The expression for the weight $\alpha_k$ in final form is

$$\alpha_k = \frac{1}{2} \ln\left(\frac{1 - e_m}{e_m}\right), \tag{16}$$

where $e_m = \frac{W_e}{W}$ is the weighted error rate of the weak classifier $h_k$.

*3.3. Step 3*

3.3.1. Step 3-1: Global Decision of the Licensed Channel Using Soft Decision Schemes

The FC combines soft energy reports of all SUs before identifying any MU for a global decision. Various soft and hard combination schemes used at the FC are equal gain combination (EGC), maximum gain combination (MGC) and majority voting as a decision criterion.

The EGC employed by the proposed method combines the individual statistical information of all SUs and assigns equal weight to each SU decision to sum coherently. The result is compared with a numerical set threshold to decide the license user spectrum as

$$G_{EGC}(i) = \begin{cases} H_1 : & \frac{1}{M} \sum_{j=1}^{M} s_j(i) \geq \gamma \\ H_0 : & otherwise \end{cases}. \tag{17}$$

The cooperative detection and false alarm probabilities $P_{d\_EGC}$ and $P_{f\_EGC}$ made by the EGC scheme based on its global decision are:

$$\begin{aligned} P_{d\_EGC} &= Pr\left\{ \frac{1}{M} \sum_{j=1}^{M} s_j(i) \geq \gamma \mid H_1 \right\}, \\ P_{f\_EGC} &= Pr\left\{ \frac{1}{M} \sum_{j=1}^{M} s_j(i) \geq \gamma \mid H_0 \right\}. \end{aligned} \tag{18}$$

In the MGC scheme, each received signal branch is multiplied by a weight proportional to the branch gain. The branches with strong signals get amplification while the branches with weak signals are further attenuated by these weights. The MGC scheme at the FC assigns higher weights to the decision of the SUs with higher SNR and low weight to the decision of SUs with low SNR as

$$G_{MGC}(i) = \begin{cases} H_1 : & \sum_{j=1, j \neq MU}^{M} \left(w_j \times s_j(i)\right) \geq \gamma \\ H_0 : & otherwise \end{cases}, \tag{19}$$

where $w_j = \frac{\eta(j)}{\sum\limits_{j=1}^{M} \eta(j)}$. The cooperative detection and false alarm probabilities of the MGC scheme are measured based on the individual sensing reports as

$$P_{d\_MGC} = \left\{ \sum_{j=1,j\neq MU}^{M} \left( w_j \times s_j(i) \right) \geq \gamma | H_1 \right\},$$
$$P_{f\_MGC} = \left\{ \sum_{j=1,j\neq MU}^{M} \left( w_j \times s_j(i) \right) \geq \gamma | H_0 \right\}. \tag{20}$$

### 3.3.2. Step 3-2: Global Decision of the Licensed Channel Using Boosted Tree Algorithm

As in the flowchart in Figure 3, the system is trained for the given number of patterns with sensing energies of the reporting users and their channel gain information as in (3) under the normal and all different categories of MUs sensing observations. At the arrival of new sensing observations from the different cooperative users that include AWGN and channel statistics the BTA decision is as follows

$$G_{BTA}(i) = \begin{cases} H_1 : & BTA\ (s^i) = 1 \\ H_0 : & otherwise \end{cases}, \tag{21}$$

where $G_{BTA}$ is the global decision of the BTA algorithm. The cooperative detection and false alarm probabilities made by the BTA algorithm at the FC are determined as

$$P_{d-BTA} = \Pr\{G_{BTA}(i) = 1|H_1\} = \Pr\{BTA\ (s^i) = 1|H_1\},$$
$$P_{f-BTA} = \Pr\{G_{BTA}(i) = 1|H_0\} = \Pr\{BTA\ (s^i) = 1|H_0\}. \tag{22}$$

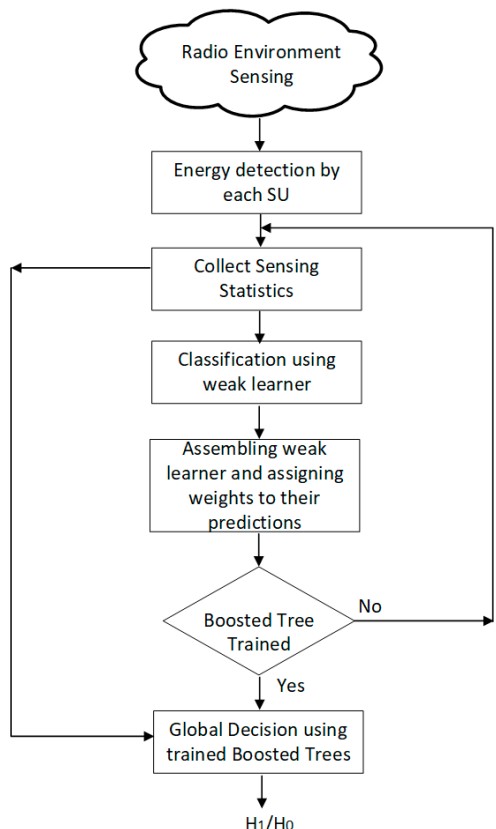

**Figure 3.** Proposed scheme flowchart.

## 4. Numerical Results

In the simulation, parameters of the CRN are adjusted with a total of 10 to 22 cooperative users. The results are simulated in the varying SNRs environment i.e., −30 dB to 0 dB. The sensing interval is selected 1 ms having 270 to 335 samples. Cooperative users placed in different SNRs sense the PU channel independently. The population size of the genetic algorithm (GA) and particle swarm optimization (PSO) consists of $M$ total genes in each $N$ chromosome. The maximum size of sensing iterations for GA and PSO schemes are kept 50. A crossover point is randomly selected in the range 1 to $M$ for the GA. Performance of the proposed BTA algorithm-based decision scheme is compared with the GASDF, PSOSDF, MGCSDF and most counting schemes.

The simulation environment is categorized into four different cases. Case I discusses the probability of error results against varying SNRs, total cooperative users and sensing samples in the presence of normal cooperative users where no MU participates in sharing data with FC. Case II shows error probability results against varying SNRs, cooperative users and sensing samples in the presence of AN and ANC categories of MUs. Similarly, Case III illustrates error probability results against varying SNRs, total cooperative users and sensing samples for all combination schemes at the participation of the AY and AYC category of MUs. The simulation results in Case IV demonstrates error probabilities under varying SNRs, cooperative users and sensing samples at the participation of AO and AOC category of MUs.

### 4.1. Case I

Here, the ABT algorithm is trained based on the sensing information reported by the normally cooperative users to perform sensing responsibilities during the testing phase. Error probabilities are collected against varying SNRs, varying cooperative users and sensing samples in Figures 4–7.

Figure 4 shows error probabilities for the proposed ABT, PSOSDF, GASDF and MGCSDF schemes against varying SNRs with a fixed number of cooperative users and total sensing samples. The proposed scheme shows similar sensing abilities to the traditional SDF schemes till −15 dB SNRs with an average of 31% less error probability, while any further increase in the SNRs values leads to improved sensing performance by the proposed ABT scheme with a 90% reduction in error probability by the ABT scheme compared with all other schemes. The proposed ABT scheme sensing results are next followed by the MGCSDF scheme in terms of error probability. Similarly, the simple CHDF shows the worst sensing performance in the normal cooperative users' environment.

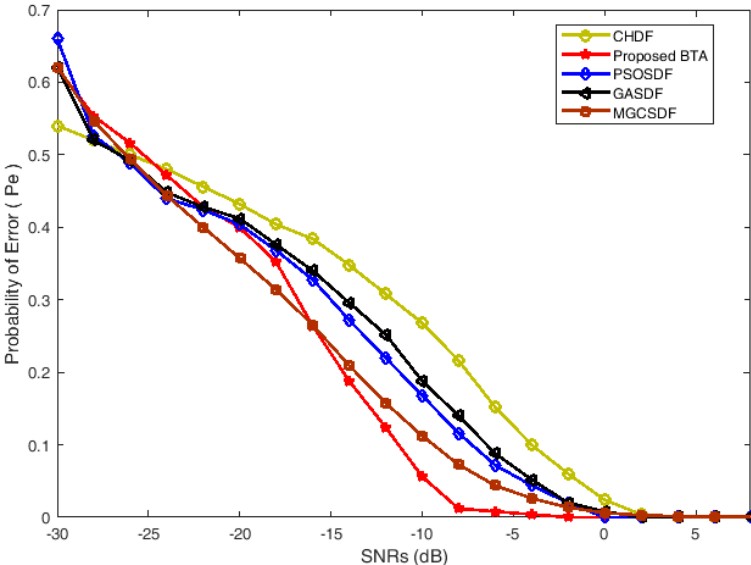

**Figure 4.** Probability of error vs. signal-to-noise ratios (SNRs) under normal cooperative users.

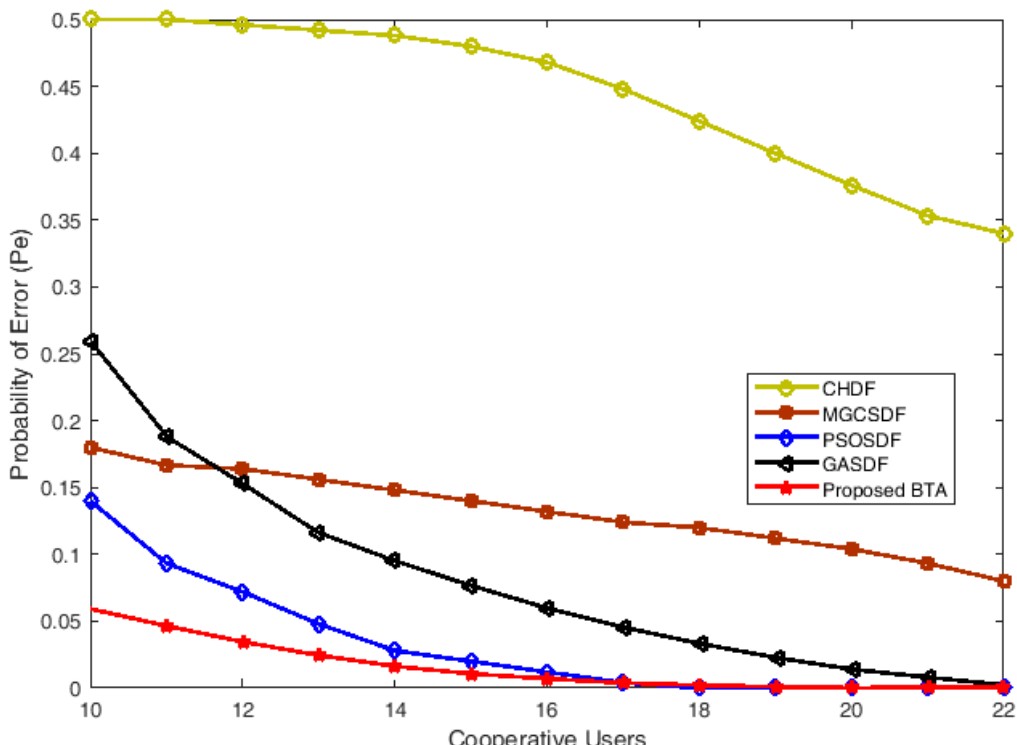

**Figure 5.** Probability of error vs. total cooperative users under normal users.

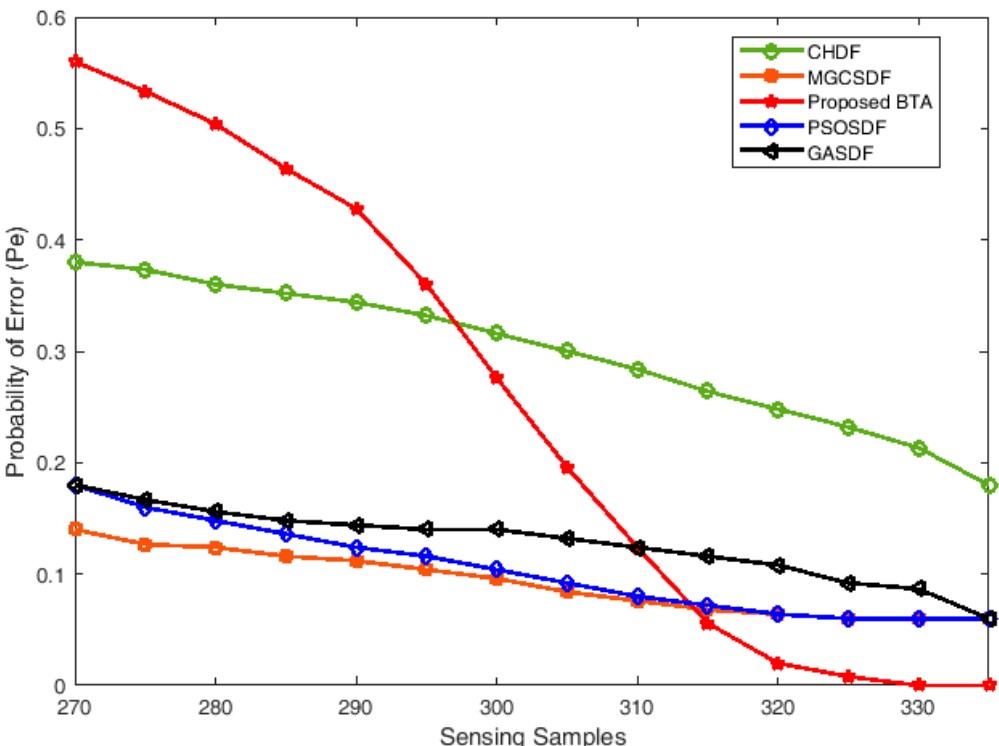

**Figure 6.** Probability of error vs. sensing samples under normal users.

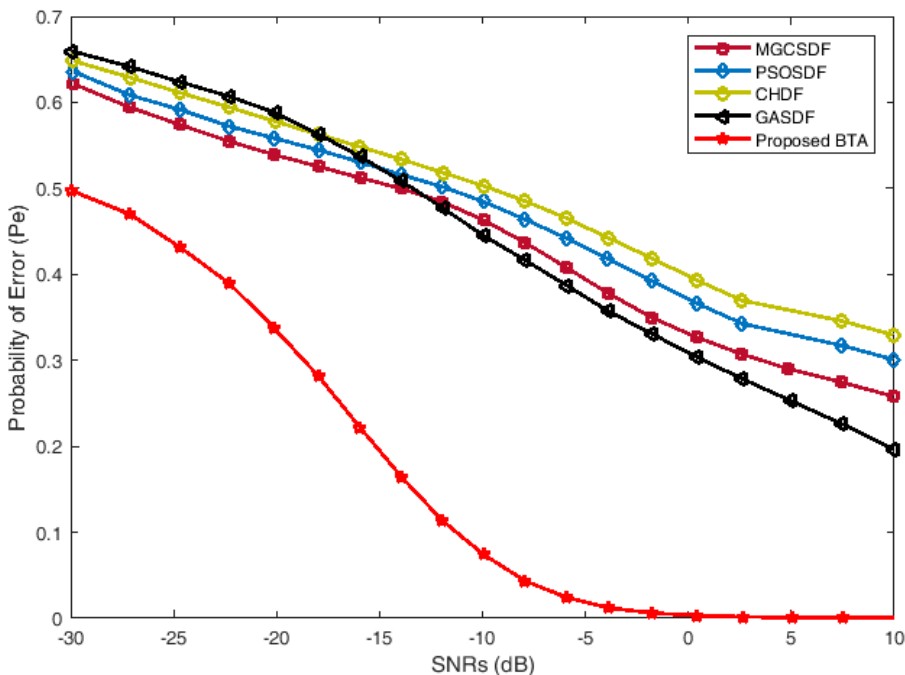

**Figure 7.** Probability of error vs. SNRs with always no category of malicious users (MUs).

Figure 5 shows error probability results against varying numbers of cooperative users without malicious users' participation. Here, the SNR and sensing samples are kept similar for all cooperative users. Results confirm better sensing performance by the proposed ABT algorithm with an average of 72% minimum error probability at M = 10, which is further reduced to 81% on the average with an increase in the number of cooperative to 15. The proposed ABT results are followed by the PSOSDF and GASDF schemes at all levels of the contributing cooperative users. The CHDF scheme has the worst sensing performance among all in the presence of total cooperative users. Similarly, Figure 6 shows error probabilities against increased sensing samples at all cooperative users in the presence of normally cooperative users. The SNR and total number of cooperative users are kept constant as SNRs = −11.5 dB and *M* = 10. Figure 6 shows with increasing sensing samples above 295 proposed ABT algorithm is able to surpass the CHDF with a 5% minimum sensing error. Similarly, an increase in sensing samples above the 315 ABT scheme is able to beat PSO-SDF, GA-SDF and MGC-SDF schemes with lower error results in sensing PU channel on the average of 43% minimum, which is further improved as the number of samples reaches 330.

*4.2. Case II*

In this case, the cooperative environment is devised by including AN and ANC categories of MUs. The AN and ANC attackers report sensing data to the FC along with the normal cooperative users. The proposed system is first trained based on the sensing reports of the normal, AN and ANC category of MUs and then error probabilities are collected against the varying SNRs, cooperative users and increased sensing samples in the testing phase.

Figure 7 shows error probabilities against increased SNRs for the proposed and traditional SDF and HDF schemes. Here, the total number of cooperative users and sensing samples are kept constant as 10 and 270. The result confirms improved sensing behavior by the proposed ABT scheme at −30 dB with an average of 22% minimum sensing error that gets further improvement to an average reduction in error probability of 96% compared with error probabilities of the other combination schemes as the SNR is increased to −5 dB. The proposed ABT algorithm is followed by the GASDF that has initially poor sensing results up to −15 dB SNRs. The PSOSDF and CHDF schemes show the worst sensing responsibilities with increasing SNRs in the presence of AN and ANC category of MUs in Figure 7.

In Figure 8 similar error probability results are collected against an increased total number of cooperative users with fixed sensing samples and SNRs values at 270 and SNRs = −11.5 dB. Figure 8 illustrated that increasing total number of normal cooperative users above 10 leads to a clear reduction in the error probability for the proposed ABT scheme as compared with PSOSDF, GASDF, MGCSDF and CHDF schemes on the average of 96% minimum sensing error compared with other combination schemes at M = 21. As in Figure 7, the GASDF is able to beat the PSOSDF and CHDF schemes above *M* by producing better sensing results with minimum sensing error.

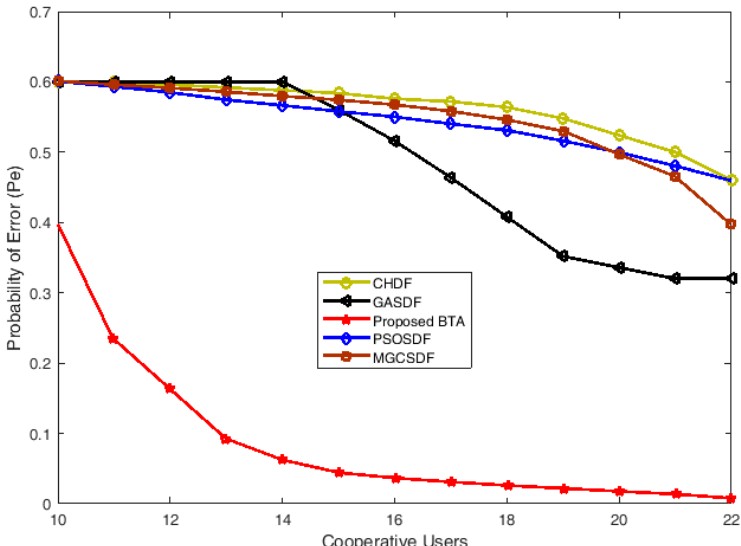

**Figure 8.** Probability of error vs. cooperative users with no category of MUs.

Figure 9 compares the proposed ABT performance against all other schemes by an increase in the number of sensing samples with fixed cooperative users *M* = 10 and SNRs = −11.5 dB. It is observable that at *K* = 305, the proposed ABT scheme is able to beat PSOSDF, GASDF and CHDF schemes. The figure shows that as the sensing samples are increased above *K* = 320, ABT is able to beat all combination schemes in terms of better sensing responsibility with minimum sensing error on the average of 100% compared with traditional techniques at *K* = 335.

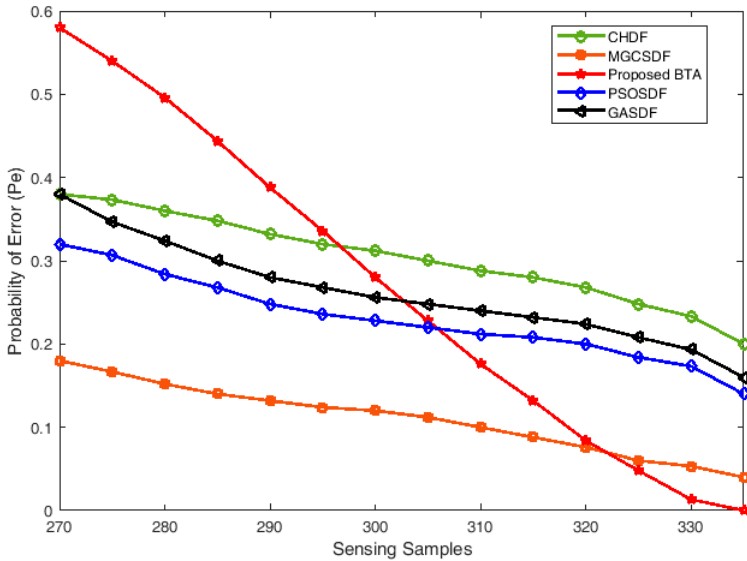

**Figure 9.** Probability of error vs. sensing samples under no users.

### 4.3. Case III

Here, the cooperative sensing performance is investigated in the presence of AY and AYC category of MUs. As the AY and AYC participants always report high energy of the PU channel to the FC, therefore the ABT scheme is trained based on the sensing reports of the normal, AY and AYC category of MU sensing data in the training phase. In the testing phase proposed ABT algorithm performance is compared with PSOSDF, GASDF, MGCSDF and CHDF schemes. Results of the error probabilities are collected in Figures 10–12.

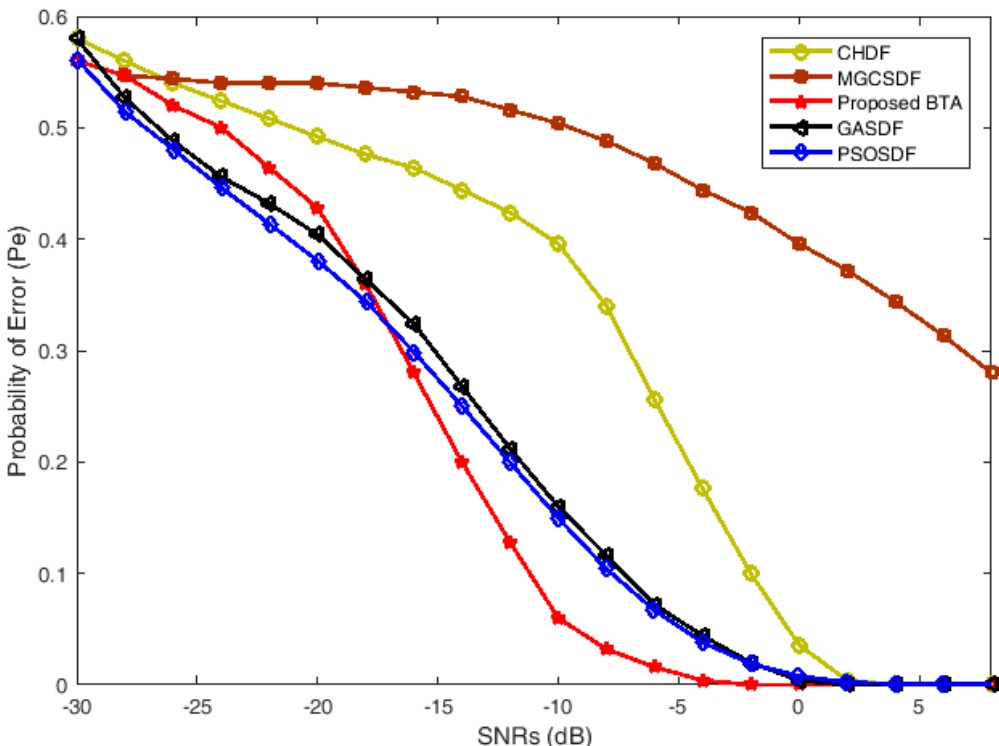

**Figure 10.** Probability of error vs. signal-to-noise ratio with the AY category of MUs.

Figure 10 shows error probabilities against varying SNRs with fixed $M = 10$ and sensing samples $K = 270$. It is clear for SNRs above $-18$ dB the proposed ABT scheme is giving sophisticated detection results with minimum sensing error on the average of 81% when SNRs reach $-8$ dB compared with all other schemes. The proposed scheme performance is next followed by the PSOSDF and GASDF schemes, while the MGCSDF scheme has the worst sensing reliability 93% higher than the proposed algorithm in this case of AY and AYC category of MUs. As the AY and AYC always report high energy statistics to the FC irrespective of the actual PU channel conditions that easily misguide the MGCSDF scheme and therefore both AY and AYC obtain high weights at the FC.

Figure 11 shows that under a minimum number of cooperative users in the presence of AY/AYC category of MUs proposed ABT scheme performance is not much reliable. Similarly, with an increasing number of cooperative users above 14, the ABT scheme is able to beat all other schemes with better sensing results of 74% minimum sensing error on the average at $M = 15$ under fixed SNR and sensing samples as SNRs = $-11.5$ dB, $K = 270$. The error probability results with varying sensing samples in the presence of AY/AYC category of MU are shown in Figure 12. The result shows that at $K = 290$, ABT is able to surpass the MGCSDF. Similarly at $K = 295$, it is producing better sensing results in comparison with the CHDF scheme and with an increase in samples above 315 the proposed scheme is beating all other combination schemes by producing minimum sensing error on the average of 100% at $K = 335$.

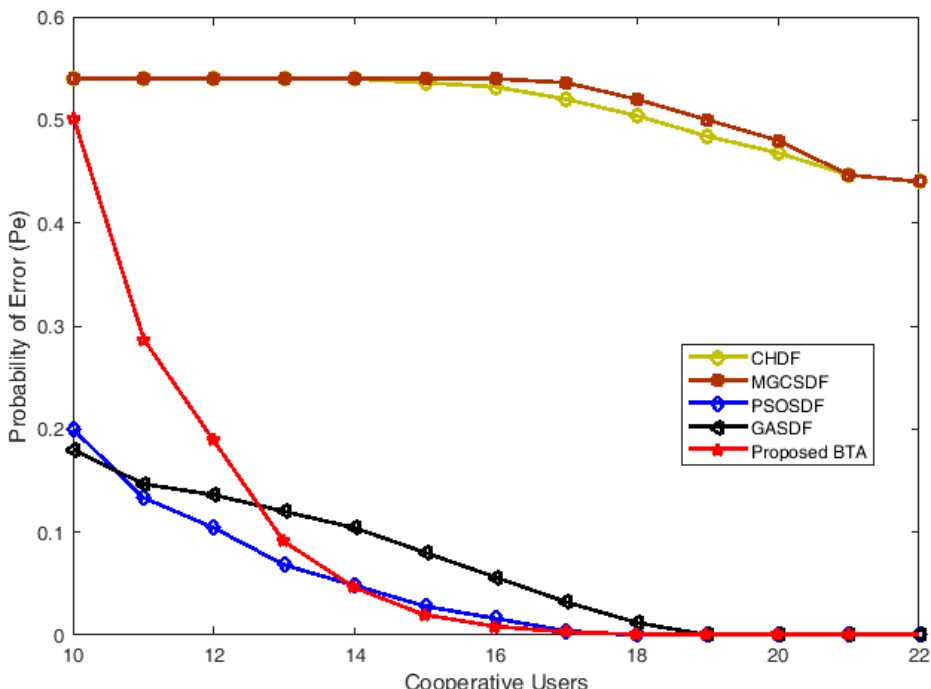

**Figure 11.** Probability of error vs. cooperative users with the AY category of MUs.

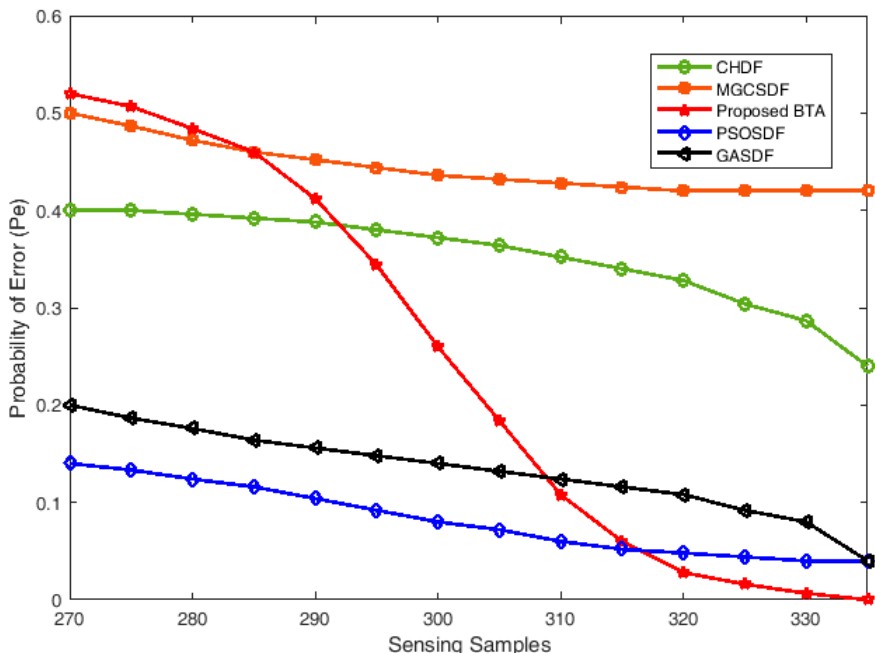

**Figure 12.** Probability of error vs. sensing samples under YES users.

### 4.4. Case IV

In the final case of the simulation results, the proposed ABT scheme performance is compared with all other schemes in the presence of AO/AOC category of MUs. The AO/AOC users report always negates the true status of the PU channel, therefore evading false reports of the RO/ROC is more critical in CSS. Here, error probability results are collected in Figures 13–15 for all combination schemes in the presence of AO/AOC category of malicious attacks. The proposed ABT scheme is trained in this case based on the sensing reports collected from the normal and the AO/AOC category of MUs in the training phase and then error probability results are collected in the testing phase.

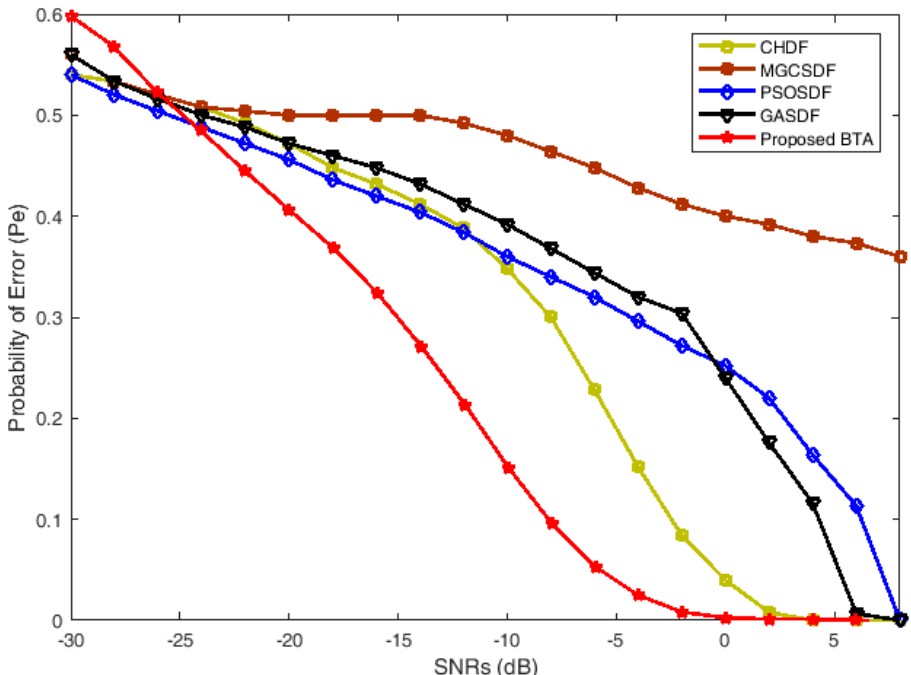

**Figure 13.** Probability of error vs. SNRs with opposite MUs.

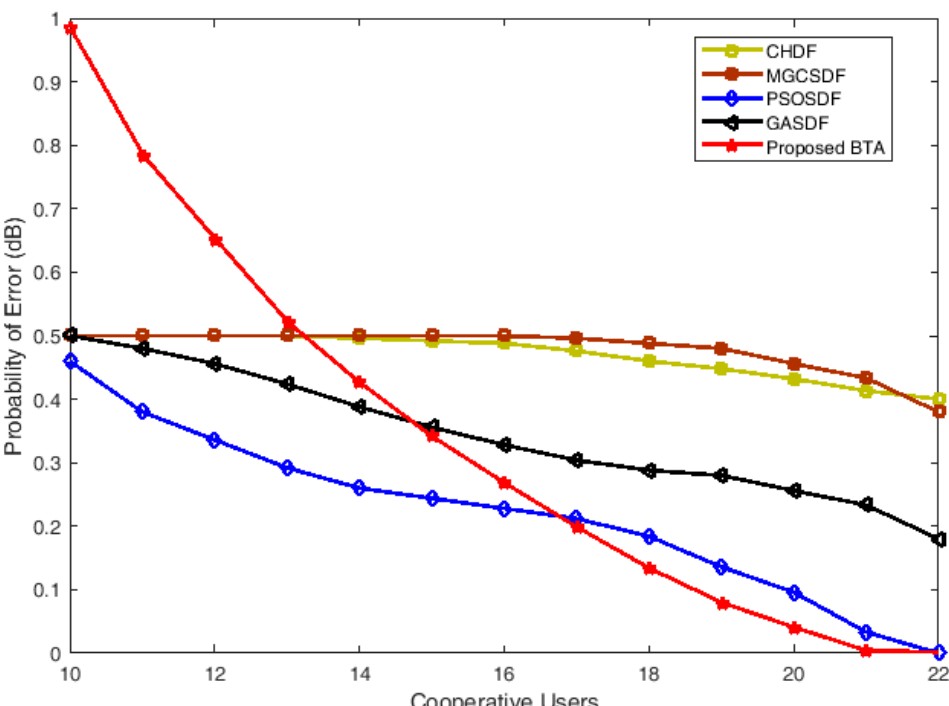

**Figure 14.** Probability of error vs. cooperative users with opposite MUs.

Figure 13 shows the probability of error against increasing SNRs and a fixed number of cooperative users $M = 10$ and sensing samples $K = 270$. The error probability results in Figure 13 shows that below $-26$ dB proposed ABT scheme has its worst sensing performance with 3% higher sensing error than the best PSOSDF scheme in this case, while further increase in SNR result in a drastic reduction in error probability for the proposed ABT scheme on the average of 73% minimum among all when

SNRs = −8 dB. Figure 13 confirms that the proposed scheme performance is next followed by the CHDF scheme, while the MGCSDF has the worst sensing abilities in this case among all.

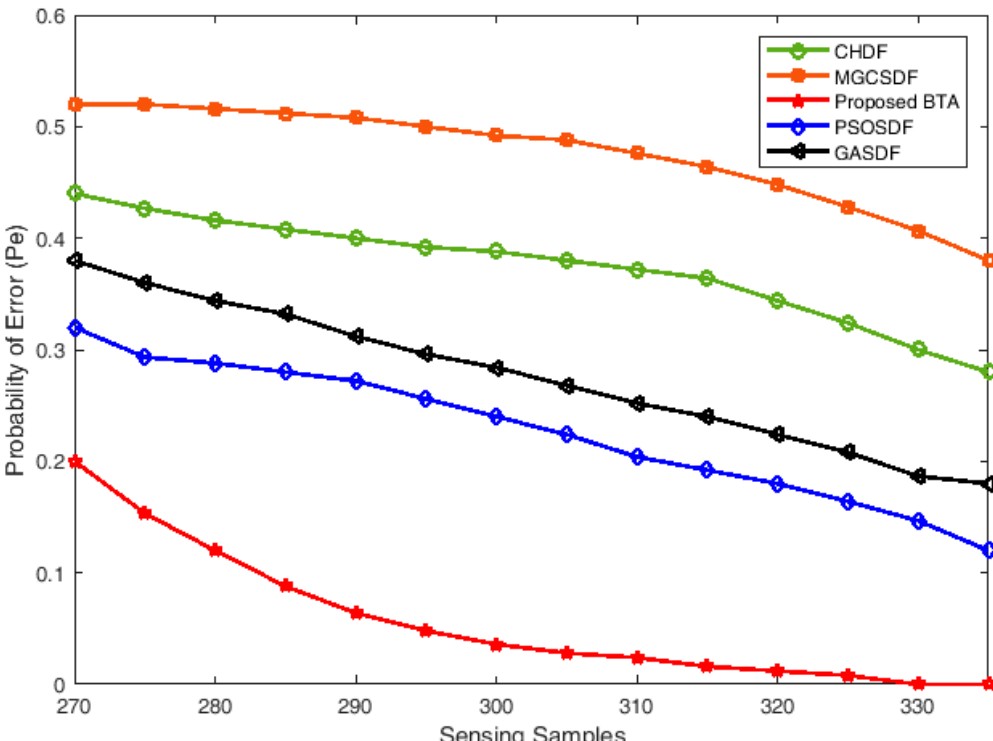

**Figure 15.** Probability of error vs. sensing samples under opposite users.

Figure 14 shows error probabilities for the proposed and all other combination schemes with an increased number of cooperative users. Here, the SNR is kept constant as SNRs = −11.5 dB and sensing samples $K = 270$. The error probability results in Figure 13 shows that as the number of cooperative users is below 13, the proposed scheme shows poor sensing performance. Similarly, as the number of cooperative users reach $M = 17$, the ABT scheme is able to surpass all other schemes on the average of 40% reduce minimum sensing error, which is further improved to an average of 70% minimum sensing error when $M = 19$. The traditional MGCSDF and CHDF schemes show their worst sensing performance among all in this case.

Figure 15 shows the error probability for the proposed and other combination schemes with increase sensing samples. The result in Figure 15 illustrates that at $K = 270$, ABT performance is comparatively much more sophisticated as compared with PSOSDF, GASDF, MGCSDF and CHDF schemes with an average of 50% minimum sensing error. Similarly, any further increases in the number of sensing samples result in a comparative reduction in the sensing error probability of the proposed scheme. The proposed scheme results are followed by the PSOSDF and GASDF schemes that produce reliable sensing results as compared with MGCSDF and CHDF schemes. The CHDF scheme in Figure 15 has the worst sensing performance in terms of sensing the PU channel among all.

## 5. Conclusions and Future Work

The fading and shadowing effects of the Rayleigh fading channel reduces the sensing performance of an individual user. Machine-learning-based CSS in this paper is able to determine suitable PU channel information using the BTA algorithm in the presence of MUs reports. The BTA scheme in the paper is compared with PSOSDF, GASDF, MGCSDF and CHDF schemes at all SNRs, sensing samples and cooperative user's participation.

Reliability of all cooperative schemes, such as BTA, PSOSDF, GASDF, MGCSDF and CHDF is investigated at the participation of AY/AYC, AN/ANC and AO/AOC categories of MUs, which shows better sensing performance by the BTASDF scheme among all. The results in this paper can be further extended in the future to check the authenticity and accuracy of these schemes with PUEA and lazy categories of MUs in CSS.

**Author Contributions:** All authors conceived and proposed the research idea. N.G. and M.S.K. designed the scenario; N.G., M.S.K. performed the simulation results; A.E., J.K. and S.M.K. Analyzed the simulation results; N.G., and Z.K. wrote the paper in the supervision of S.M.K. and J.K. All authors have read and agreed to the published version of the manuscript.

**Funding:** This work was supported in part by the MIST (Ministry of Science and ICT), Korea under the ITRC (Information Technology Research Center) support program (IITP-2020-2018-0-01426) supervised by the IITP (Institute for Information and Communication Technology Planning & Evaluation) and in part by the National Research Foundation (NRF) funded by the Korea government (MSIT) (No. 2019R1F1A1059125).

**Conflicts of Interest:** The authors declare no conflicts of interest.

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
