# Peer review of "Boosted Trees Algorithm as Reliable Spectrum Sensing Scheme in the Presence of Malicious Users"

_electronics, doi:10.3390/electronics9061038_

Round 1
Reviewer 1 Report
In this paper, the authors used a boosted tree algorithm (BTA) for obtaining reliable identification of the primary users (PU) channel, where the secondary users (SUs) can access the PU channel opportunistically with minimum disturbances to the licensee. Authors claim that the proposed BTA mitigates the spectrum sensing data falsification (SSDF) effects of the Always Yes (AY), Always No (AN), Always Opposite (AO), and Random Opposite (RO) categories of the malicious users (MUs). My primary comment will be the lack of novelty in the algorithm or in the study. The authors failed to explain the novelty of the proposed algorithms on how it is different from existing BTA models.
Authors need to provide better clarity in the introduction on how they come to the decision to use the BTA algorithm? Instead of Random Forest, or Decision Tree, Deep Neural Network, or Data Fusion, or Bayesian models? What kind of review is done for literature? etc. These are important questions to be answered to understand the novelty of the proposed BTA.
The following manuscript may help authors:
- Performance Evaluation of Distributed Compressed Wideband Sensing for Cognitive Radio Networks
- Data Fusion Schemes for Cooperative Spectrum Sensing in Cognitive Radio Networks
- Defending Against Cooperative Attacks in Cooperative Spectrum Sensing
- Spectrum Sensing for Cognitive Radios: Algorithms, Performance,
and Limitations - Trusted Collaborative Spectrum Sensing for Mobile Cognitive Radio Networks
- Byzantine Attack and Defense in Cognitive Radio Networks: A Survey
The system model is missing proper reference, on what basis authors modeled the system presented in section 2? Present proper references.
The presentation of the manuscript can be improved I suggest redrawing figure 1, and adjusting resolutions of figure 2 and figure 3.
Many of the mathematics used in section 3 proposed a boosted tree algorithm, is in many previous published literature, considered reproducing published work. I recommend only using necessary formulation that is required for proposed application.
Authors claimed that the proposed BTA is better sensing in comparison with all the models they have compared such as PSOSDF, GASDF, MGCSDF, and CHDF. This is not supported by section 4 results, results in section 4 are highly worrying showing BTA performance all over the places with no strong dominance. I consider this is not significantly better performance, to consider this model as better performance authors need to show how much better the proposed BTA algorithm performed.
The manuscript has potential for an application-oriented paper, I recommend authors to improvise the manuscript including results and resubmit.
Reviewer 2 Report
The submitted manuscript is good and looks novel. Authors proposed an algorithm using boosted trees algorithm to improve the security of the cooperative spectrum sensing by mitigating the impact of the malicious users. The authors provided a mathematical analysis of the proposed algorithm and compared its simulation results with well-known existing algorithms. The comparison shows the effectiveness of the proposed algorithm. However, I have some major and minor notes which I strongly recommend the authors to address and consider them to add some more quality to their submitted work.
Major notes:
- In the section of the simulation results, the authors just narrated the behaviour of the proposed algorithm without discussing the reasons behaviour. In other words, why the proposed algorithm outperforms the other algorithm in some cases and at specific sensing samples. what is the penalty that the proposed algorithm will pay for this improvement?
- The introduction referred to outdated references, I highly recommend authors to refer to recent reference such as
- "Detection of Adversary Nodes in Machine-To-Machine Communication Using Machine Learning-Based Trust Model." 2019 IEEE International Symposium on Signal Processing and Information Technology (ISSPIT). IEEE, 2019.
- "Security and Privacy Protection for Mobile Applications and Platforms." Mobile Networks and Applications (2020): 1-2.
- "Blind spectrum sensing approaches for interweaved cognitive radio system: A tutorial and short course." IEEE Communications Surveys & Tutorials 21.1 (2019): 238-259.
- "Technical issues on cognitive radio-based Internet of Things systems: A survey." IEEE Access 7 (2019): 97887-97908.
Minor notes
- eliminate big spaces on pages 4 , 11.
- define MU on page 2 in the section of the introduction
- cyclo-stationary detector is a type of feature detector ( page 2)
- P2, line 88 add space " proposedin [14]"
- The manuscript needs to check punctuation.
- the authors should be consistent in writing their manuscript, sometimes they used simple past tense and at the same time they simple present tense.
- P 6, L 206 and 215 replace in() by where.
- P 7, from L249 to L 257 many square brackets are missing.
- Fig 3 make it clear.
- P 15, L 406, add spaces for " SNRs=-11.5".
- Please try to separate the figures and their explanation. Usually figures are put on the top of the page and their explanation comes below. do not mix please.
Finally, I wish all the best for the authors.
Round 2
Reviewer 1 Report
The authors responded to reviewers' comments in detail and improved the results and presentation. Based on the new improvements and additions I recommend publication of the manuscript with minor improvements and final revisions to English and grammar. I recommend authors to review and edit English and flow of the paper one more time before publication.
Reviewer 2 Report
Thanks so much for considering and addressing my notes, I wish you best of luck.